# Sex, Sleep Duration, and the Association of Cognition: Findings from the China Health and Retirement Longitudinal Study

**DOI:** 10.3390/ijerph181910140

**Published:** 2021-09-27

**Authors:** Wei Li, Ning Sun, Anthony Kondracki, Wenjie Sun

**Affiliations:** 1Department of Epidemiology, Robert Stempel College of Public Health, Florida International University, Miami, FL 33199, USA; wli034@fiu.edu (W.L.); nsun004@fiu.edu (N.S.); 2Department of Community Medicine, School of Medicine, Mercer University, Savannah, GA 31404, USA; akondracki2012@gmail.com; 3School of Public Health and Tropical Medicine, Tulane University, New Orleans, LA 70112, USA

**Keywords:** sleep duration, sex difference, cognition impairment, aging Chinese

## Abstract

Background: We aimed to examine the association between sleep duration and cognitive impairment among elderly Chinese people. Methods: generalized linear modeling was used to analyze the baseline data for adults aged 65 years and over (*n* = 4785) selected from the 2011 China Health and Retirement Longitudinal Study (CHARLS). The two aspects of cognitive impairment assessed in the study were mental status and memory. Sex-stratified logistic regression models were conducted to identify the effect of sleep duration in the different gender groups. Results: of all the participants, 49.8% were females and 32.5% aged 75 years and over. Of the participants, 59.7% had short sleep duration (<7 h/day), and 9.0% of them had long sleep duration (>8 h/day). Compared to normal sleep duration, long sleep duration was associated with lower mental status scores (β = −0.43, *p* = 0.001) and lower memory scores (β = −0.26, *p* = 0.006). Long sleep duration was associated with lower mental status in both men (β = −0.37, *p* = 0.033) and women (β = −0.46, *p* = 0.025), however, only in men was long sleep duration found to be associated with low memory scores (β = −0.26, *p* = 0.047). Conclusions: Our study showed that long sleep duration was significantly associated with poorer mental status and memory scores in elderly Chinese people. Paying greater attention to the effects of sleep patterns on the risk of cognitive decline may yield practical implications for dementia prevention and health promotion, especially among older women with lower educational attainment, living in rural areas, and those who have long sleep duration.

## 1. Introduction

The percentage of elderly in the world’s population is rapidly increasing. It has been projected that between 2015 and 2050, the proportion of the world’s population that is 60 years of age and older will nearly double, from 12% to about 22% [1]. China, the most populated country in the world, has a high ratio of seniors (aged 60 and above) to working-age adults [2]. Some of the serious health concerns for older adults, globally, include cognitive decline and symptoms of dementia related to the aging brain [3,4,5]. Dementia describes a chronic or progressive reduction in cognitive function affecting memory, thinking and social abilities [6]. Cognitive impairment has a significant impact on senior health and the activities of daily living, from issues related to safety and self-care, to the cost burden of long-term care for families and society [7,8]. Because no effective cure currently exists for dementia, research is trying to identify potential risk factors affecting brain health to help halt the progression of dementia [3,6]. 

There is growing interest in the interaction between aging and sleep-related problems that commonly include a decrease in total sleep duration and efficiency, an increase in sleep fragmentation and sleep disturbances, as well as greater difficulty falling asleep [7,8,9]. Accumulated evidence shows that sleep duration can be associated with an increased risk of dementia in older adults, though the results are inconsistent [4,10,11,12]. For example, in a recent meta-analysis study, Liang et al. reported that both short and long sleep duration could be associated with an increased risk of dementia [12], whereas, findings from a 25-year cohort study indicated that only short sleep duration (vs. normal sleep duration) was associated with a 30% increase in dementia risk among older adults aged over 50 years [4]. The diagnosis of dementia is based on the evaluation of impairment in the activities of daily living, memory, language, attention, and visuospatial cognition [13]. Thus, understanding the relationship between sleep duration and cognitive decline is of great significance for dementia prevention. 

Accordingly, the aspects of cognitive function and sleep patterns are slightly different between men and women, while the results are inconsistent [14,15]. For instance, one study found that sleep loss could impair working memory in women, but not in men [14]. The possible explanation was that, compared to men, women exhibited greater signal intensity changes when performing a working memory task and were more vulnerable to sleep loss [14]. However, another study suggested that certain aspects of cognition (e.g., sustained attention) may be more affected by sleep loss (38 h of sleep deprivation) in men than in women due to the sex differences in brain organization and anatomy [15]. Therefore, examining the sex-specific associations between sleep duration and cognitive function may provide further mechanistic insight and allow for additional tailored interventions for dementia prevention. In the presented study, we used data from the China Health and Retirement Longitudinal Study (CHARLS), a nationally representative aging cohort that provided a large sample size and included cognitive assessments [16] to examine the association of sleep duration with cognitive impairment among elderly Chinese people, as well as the sex differences between them. We assumed that (1) long time or short time sleep duration could be associated with poor cognition and (2) there might be a sex difference in the sleep-cognition relationship.

## 2. Materials and Methods

### 2.1. Study Sample

This cross-sectional study included data on 4785 adults aged over 65 obtained from the China Health and Retirement Longitudinal Study (CHARLS). CHARLS is a national population-based survey of community-dwelling adults aged 45 years and older including 10,287 households, 17,708 individuals, and covering 150 counties in 28 provinces, and gathering information with a focus on the health and socioeconomic status of the rapidly aging population [16]. The first wave was conducted between June 2011 and March 2012 and the participants were followed up biennially until wave 4 in 2018, using a face-to-face computer-assisted personal interview (CAPI) [16]. A multistage stratified probability proportional to size sampling was adopted in CHARLS. More details regarding CHARLS are described elsewhere [16]. We only selected elderly people who were aged 65 years and older since different age groups have different recommended sleep duration according to the National Sleep Foundation (NSF) guidelines [17]. Ethics approval for the CHARLS study was obtained from the Institutional Review Board (IRB) at Peking University. The IRB approval number for the main household survey was IRB00001052-11015 and for the biomarker collection was IRB00001052-11014.

### 2.2. Outcome

Cognitive function was captured using two measures guided by previous literature [18,19]: the mental status score and the memory score. The mental status score includes questions on orientation (the ability to identify today’s date, season, day of the week), visuoconstruction (the ability to redraw a previously shown picture), numeric ability (subtract 7 five consecutive times), and whether additional explanation or aid was needed. The total score ranged from 0 to 11, with a higher score indicating better cognitive function. The questions were similar to the Telephone Interview of Cognitive Status (TICS-10) [19]. The memory score was measured by immediate and delayed recall of 10 Chinese words that were read to the respondents [18]. The total score ranged from 0 to 10, calculated by averaging the immediate and delayed recall scores. Higher scores indicated better mental health and memory function. 

### 2.3. Exposure and Covariates

Sleep duration was an exposure categorized into three groups: short sleep (<7 h), normal sleep (7–8 h), and long sleep (>8 h), according to the National Sleep Foundation (NSF) [17]. It was evaluated by the following question: “During the past month, how many hours of actual sleep did you get at night (average hours for one night)?”

Covariates were selected based on previous literature [3,4,5,20] and included sociodemographic characteristics (age in years, biological sex, residence (urban/rural), marital status, education level, and insurance status), lifestyle variables (physical activity, napping (Yes/No), smoking, and drinking) comorbidity and health status (self-rated health, memory-related disease, and depression). Physical activity was assessed by asking “During a usual week, did you do any vigorous/moderate/light activities for at least 10 min continuously?” If the participants answered “No” to all of them then they were grouped into insufficient activity. Smoking status was classified as current smoker, former smoker, and nonsmoker. Drinking status was categorized as current low frequency use (<5 drinks per month), current high frequency use (≥5 drinks per month), former use (quit drinking) and never use, based on the Substance Abuse and Mental Health Services Administration (SAMHSA) categories [21]. Self-rated health was assessed by the following question “Would you say your health is excellent, very good, good, fair, poor or very poor?” and then coded as good (including excellent, very good and good) and fair/poor (including fair, poor and very poor). Memory-related disease used the question “Have you been diagnosed with memory-related disease by a doctor?” Depression was assessed with the CES-D-10 scale (range 0–30), and then recoded as a binary variable with the cutoff score of 11 [22]. Employment status was classified as farm (engaged in agricultural work for more than 10 days in the past year), work (worked for at least one hour last week) and retired (neither of the two options above). 

### 2.4. Statistical Analysis

Statistical analysis was performed with SAS 9.4 (SAS Institute, Cary, NC, USA). Characteristics of the overall sample and of each sleep duration group were described with means/standard deviations for continuous variables and counts/percentages for categorical variables. Chi-squared tests and the one-way analysis of variance (ANOVA) were used to examine the association between sleep duration groups and the characteristic variables. Sex-stratified generalized linear models, controlled for sociodemographic characteristics, lifestyle variables, comorbidity and health status were used to evaluate the relationship between sleep duration and cognition. Estimates and standard errors (SE) of the coefficients (β) were reported. The significance level was defined as a 2-sided alpha value of 0.05.

## 3. Results

Table 1 shows the characteristics of the overall sample. A total of 49.8% were female and 32.5% aged 75 years and over. More than half of the participants had an education level less than elementary (60.9%), were married and lived with spouse present (68.7%), lived in rural area (76%), had insurance (92.9%). More detailed information is provided in Table 1. Additionally, 59.7% of the participants slept less than 7 h/day, and 9% of them slept longer than 8 h/day. Participants who slept 7–8 h/day had the highest mental status scores (6.82 ± 2.1) and memory scores (3.14 ± 1.7), and those with long sleep duration had the lowest mental status scores (5.99 ± 2.2) and memory scores (2.65 ± 1.6).

As shown in Table 2, for the overall population, long sleep duration (>8 h per day) was associated with 0.43 points lower in mental status scores (β = −0.43, *p* = 0.001), compared to individuals with normal sleep duration (7–8 h per day). Similarly, compared to individuals with normal sleep duration, long sleep duration was associated with 0.26 points lower in memory scores (β = −0.26, *p* = 0.006). No significant association was found between short sleep duration and cognition scores.

When stratified by sex, a slight difference was noticed between male and female responders regarding the associations mentioned above. In men, compared to normal sleep duration, long sleep duration was associated with 0.37 points lower (β = −0.37, *p* = 0.033) and 0.26 points lower (β = −0.26, *p* = 0.047) in mental status scores and memory scores, respectively. However, in women, compared to normal sleep duration, the association was significant only between long sleep duration and mental status scores (β = −0.46, *p* = 0.025). 

In addition, living in an urban area and higher educational attainment were associated with higher mental status scores and memory scores for both male and female participants. Older age and depression were associated with lower mental status scores and memory scores for both men and women. Participating in agricultural production was associated with higher memory scores. Marital status (married with vs. never married) was associated with higher mental status scores for men only (β = 0.84, *p* = 0.025). Self-rated health as fair/poor (vs. good) was associated with lower memory scores for men only (β = −0.22, *p* = 0.012).

## 4. Discussion

The results of our study indicate that long sleep duration was associated with poorer cognitive performance, measured by mental status scores and memory scores, compared to normal sleep duration. Moreover, long sleep duration was associated with poorer cognitive performance in mental status scores for the overall population. However, the association between long sleep duration and poor cognitive performance in memory scores was found in males only. These findings provided significant insights, especially for guiding the sex-specific interventions for cognition dysfunctions and dementia prevention.

The relationship between sleep and cognition is complex and poorly understood. Most past sleep studies were conducted in laboratories with strict conditions. Long sleep duration was associated with better cognitive function among children [23], while this relationship was not clear among the elderly [24]. A previous study pointed out that both short and long sleep duration could be associated with worse cognitive outcomes, as indicated by a U-shape or nonlinear relationship between sleep duration and cognition [3,12]. The latter study also demonstrated that sleep duration might have a small negative effect on reaction time and visual memory [3]. Moreover, according to a national study of US older adults, long sleep duration may be a marker of fragmented sleep or neurodegeneration [25]. Our study observed a significant relationship between long sleep duration and worse performance on cognition, although the association between short sleep duration and the cognition outcomes (i.e., mental health and memory decline) was not statistically significant. The biological mechanism could be listed as follows. First, sleep duration has been potentially linked with sleep quality [3,26]. Poor sleep quality or sleep deprivation can disrupt the circadian rhythm regulated by gene expression in the frontal, thalamic, hypothalamic, and the locus coeruleus areas of the brain, causing a disturbance in neurogenesis [27] and hippocampal function [28], thus resulting in cognitive decline. Sleep-related disorders may have different effects on brain function linked to specific cognitive domains such as the synchronization function of the prefrontal cortex and neuromodulator system in visual memory [29]. Second, the relationship between sleep and cognitive function could be partly explained by the hormone melatonin which plays a critical role in sleep and cognitive performance, especially among the elderly [30,31]. Further, the alternative explanation is that circadian factors interact to influence cognitive performance. Long sleep duration may contribute circadian disruptions linked with sleep dysregulation and impaired cognition [32,33]. Therefore, long sleep duration may represent a potential causal pathway for cognitive dysfunction. 

Long sleep duration is known to be related to mental health issues such as depression [34]. Our study findings that long sleep duration was associated with lower memory scores are in line with a recent study by Kondo et al. [35]. Another possible reason behind this may be the effect of sleep duration on heightened systemic inflammation and a subsequent increase in serum β-amyloid levels [36]. Unlike short sleep duration, long sleep duration has been linked to increased C-reactive protein and interleukin-6 levels [37]. Moreover, according to the existing literature, impaired sleep quality resulting from sleep fragmentation and hypoxia may be associated with a build-up of amyloid plaques in the brainstem along with the tau phosphorylation characteristic of Alzheimer’s disease [20]. Therefore, the association between long sleep duration and the impairment of logical memory may be mediated by inflammation and β-amyloid burden.

Older age was also associated with poorer mental status and memory scores for both men and women. It could be partly explained by the impaired function of the hippocampus that is associated with learning and memory [38]. Additionally, functional impairment and reduced cognitive performance have been associated with a reduction in autophagy-related proteins in the brain [38]. Furthermore, we found that living in an urban area and higher educational attainment were associated with better mental status and memory scores. The brain adapts in response to challenges, and it seems that cognitive skills may be improved by challenging the brain with intellectually stimulating activities [39,40]. Moreover, education inequalities between rural and urban areas in China have been increasing in the last decades [41]. In contrast to rural areas, people residing in urban areas commonly have higher levels of educational attainment [41]. Therefore, elderly people who live in urban areas and have higher educational attainment are less likely to exhibit cognitive dysfunction.

We also found sex disparities regarding the associations between sleep duration and cognitive performance. Specifically, women were more likely than men to report a better (higher) mental status score of cognitive performance. It was suggested that these differences may be explained by China’s current economic resources and Chinese culture, where the preference for male children is motivated by economic and social desire to provide financial support [18]. Therefore, it is likely that the relationship between sleep duration and cognitive performance in the presented study may be influenced in a similar way. Based on previous literature, divorced and widowed marital status was also associated with poor cognitive performance [42]; however, findings from our study were inconsistent. We did not find a significant association between the divorced and widowed elders and the cognitive function, but there was a significant association between married men not living with their spouses and a higher mental status score from Table 2. A possible explanation for this may be that men obtain greater health and cognitive benefits from marriage than women. For example, in traditional Chinese marriages, a woman is responsible for social connections between friends and family, and also offers physical and emotional support for her spouse. Consequently, among married men, the risk of cognition is reduced and the cognitive benefits they gain may have a positive influence on their health [42].

Moreover, a previous systematic review and meta-analysis showed that depression was significantly associated with most of the identified cognitive variables including working memory and long-term memory [43], which is in agreement with the findings in our study indicating that depression is associated with poor mental status and lower memory scores for both males and females. This highlights the importance of treatment for depression and more research into effective interventions targeting depression and cognitive dysfunction in elderly people in China. Research explores the underlying neuroprotective mechanisms of regular physical activity to enhance overall brain health and improve cognitive function [44]. From Table 2, we found that doing moderate physical activity at least 10 min continuously a week (vs. insufficient physical activity) was marginally associated with a better memory score for males only. Future longitudinal studies with a better study design (e.g., a cohort study or panel study) are warranted to investigate these associations in men and women. 

The presented study has several limitations. First, our findings were based on a cross-sectional design and causality cannot be established between sleep duration and cognition. Second, self-reported responses (e.g., sleep duration, cognitive impairments) may either underestimate or overestimate the distribution in each variable’s categories, resulting in inaccurate assessment of the relationship between sleep duration and cognition. Third, more than half of the individuals had low educational attainment, which may bias the results since higher educational level is always associated with better cognitive scores. Further, we noticed the cognitive function score for mental health was above the average (i.e., over 6, indicating a relatively higher score), which might bias the results. Lastly, even though we tried to adjust nap and sleep disturbance as sleep pattern domain, another important factor such as sleep quality was not included because it was not available in the CHARLS dataset. These factors are extremely important and future studies need to account for these potential confounders to better assess the association between sleep duration and cognition. Despite its limitations, this study based on national representative cohort data provides novel evidence that long sleep duration is related to poorer cognitive scores in the elderly Chinese population, especially those aged 65 years and older. The study findings may inform public health stakeholders and may be used in targeted interventions to prevent cognitive impairment and dementia progress among elderly people, especially those who are less educated, live in rural areas, and have long sleep duration. Such knowledge may help inform guidelines about sleep duration to improve cognitive and overall health among seniors. 

## 5. Conclusions

This study showed that long sleep duration was significantly associated with poorer mental status and memory scores in both men and women, while the association between long sleep duration and poorer memory scores was found only in men after stratifying by sex. Greater research attention on cognition might yield practical implications for dementia prevention and health promotion for the older adult population in China, especially for older women with less education and living in rural areas who have long sleep duration. Further longitudinal studies are warranted to establish causality regarding the sleep-cognition relationship.

## Figures and Tables

**Table 1 ijerph-18-10140-t001:** Basic characteristics related to sleep duration among Chinese people aged over 65 years old (*n* = 4785).

Variables		Sleep Duration (h/day)			
	Total/Overall	<7	7–8	>8	*p*-Value *
Total	4785	2859 (59.7)	1496 (31.3)	430 (9.0)	
Sociodemographic characteristics					
Age					0.0003
65–70	1853 (38.7)	1075 (37.6)	630 (42.1)	148 (34.4)	
70–75	1375 (28.7)	800 (28.0)	444 (29.7)	131 (30.5)	
≥75	1557 (32.5)	984 (34.4)	422 (28.2)	151 (35.1)	
Biological sex (female)	2383 (49.8)	1517 (53.1)	660 (44.2)	206 (47.9)	<0.0001
Education Less than elementary Elementary Middle school More than high school	2903 (60.9) 1019 (21.4) 482 (10.1) 367 (7.7)	1776 (62.4) 576 (20.2) 301 (10.6) 193 (6.8)	834 (55.8) 350 (23.4) 155 (10.4) 156 (10.4)	293 (68.1) 93 (21.6) 26 (6.1) 18 (4.2)	<0.0001
Marital status Never Married with spouse present Married not living with spouse Separated/divorced/widowed	41 (0.9) 3287 (68.7) 128 (2.7) 1367 (28.6)	25 (0.9) 91 (3.2) 1890 (66.2) 850 (29.8)	12 (0.8) 28 (1.9) 1108 (74.1) 348 (23.3)	4 (0.9) 9 (2.1) 289 (67.2) 128 (29.8)	<0.0001
Residence (urban)	1146 (24.0)	665 (23.3)	412 (27.6)	69 (16.1)	<0.0001
Insurance (yes)	4372 (92.9)	2592 (92.7)	1394 (93.9)	386 (90.4)	0.0360
Employment status					
Farm	1781 (37.7)	1030 (36.7)	587 (39.4)	164 (38.2)	0.0879
Work	147 (3.1)	76 (2.7)	54 (3.6)	17 (4.0)	
Retired	2801 (59.2)	1704 (60.6)	849 (57.0)	248 (57.8)	
Lifestyle					
Smoking status Current Former Never	1282 (27.9) 580 (12.6) 2732 (59.5)	687 (25.7) 306 (11.5) 1677 (62.8)	475 (31.8) 215 (14.4) 804 (53.8)	120 (27.9) 59 (13.7) 251 (58.4)	<0.0001
Drinking status Current high frequency Current low frequency Former Never	483 (11.2) 535 (12.4) 394 (9.1) 2914 (67.4)	277 (10.8) 298 (11.6) 220 (8.6) 1765 (69.0)	160 (11.7) 184 (13.4) 129 (9.4) 897 (65.5)	46 (11.6) 53 (13.4) 45 (11.4) 252 (63.6)	0.1772
Physical activity Insufficient Light Moderate Vigorous	630 (13.2) 506 (10.6) 398 (8.3) 3251 (67.9)	2002 (70.0) 349 (12.2) 272 (9.5) 236 (8.3)	956 (63.9) 220 (14.7) 185 (12.4) 135 (9.0)	293 (68.1) 61 (14.2) 49 (11.4) 27 (6.3)	<0.0001
Daytime napping (yes)	2478 (56.5)	1292 (52.5)	937 (62.7)	249 (58.2)	<0.0001
Comorbidity and health status Cognitive function (mean ± SD) Mental status score Memory score	6.50 ± 2.2 2.99 ± 1.6	6.37 ± 2.3 2.95 ± 1.6	6.82 ± 2.1 3.14 ± 1.7	5.99 ± 2.2 2.65 ± 1.6	<0.0001 <0.0001
Depression (yes)	1526 (31.9)	1043 (36.5)	365 (24.4)	118 (27.4)	<0.0001
Memory-related diseases (yes)	171 (3.6)	119 (4.2)	36 (2.4)	16 (3.7)	0.0095
Self-rated health Good Fair/poor	962 (20.3) 3770 (79.7)	443 (15.8) 2366 (84.2)	402 (26.9) 1091 (73.1)	117 (27.2) 313 (72.8)	<0.0001

******p*-values for differences between sleep duration groups were calculated by Chi-square tests for categorical variables.

**Table 2 ijerph-18-10140-t002:** Generalized linear modeling assessing the association between sleep duration and cognitive function (*n* = 4785).

Variables	Overall				Male (*n* = 2401)				Female (*n* = 2383)			
	Mental Status Score *		Memory Score *		Mental Status Score *		Memory Score *		Mental Status Score *		Memory Score *	
	Estimate β (SE)	*p*-Value	Estimate β (SE)	*p*-Value	Estimate β (SE)	*p*-Value	Estimate β (SE)	*p*-Value	Estimate β (SE)	*p*-Value	Estimate β (SE)	*p*-Value
Sleep duration (hours/day)												
7–8	Ref		Ref		Ref		Ref		Ref		Ref	
<7	−0.12 (0.08)	0.119	−0.02 (0.06)	0.782	−0.18 (0.1)	0.078	0.07 (0.08)	0.357	−0.05 (0.12)	0.664	−0.1 (0.08)	0.216
>8	−0.43 (0.13)	0.001	−0.26 (0.10)	0.006	−0.37 (0.17)	0.033	−0.26 (0.13)	0.047	−0.46 (0.2)	0.025	−0.25 (0.14)	0.074
Age 65–70	Ref		Ref		Ref		Ref		Ref		Ref	
70–75	−0.32 (0.09)	<0.001	−0.34 (0.06)	<0.001	−0.27 (0.11)	0.016	−0.33 (0.09)	<0.001	−0.39 (0.13)	0.003	−0.36 (0.09)	<0.001
≥75	−0.77 (0.10)	<0.001	−0.72 (0.07)	<0.001	−0.60 (0.13)	<0.001	−0.71 (0.1)	<0.001	−0.94 (0.14)	<0.001	−0.72 (0.1)	<0.001
Biological sex												
Female	Ref		Ref		-	-	-	-	-	-	-	-
Male	0.88 (0.10)	<0.001	−0.02 (0.07)	0.745	-	-	-	-	-	-	-	-
Education												
Less than elementary	Ref		Ref		Ref		Ref		Ref		Ref	
Elementary	1.21 (0.09)	<0.001	0.56 (0.07)	<0.001	1.00 (0.12)	<0.001	0.54 (0.09)	<0.001	1.41 (0.15)	<0.001	0.60 (0.11)	<0.001
Middle school	1.43 (0.12)	<0.001	0.80 (0.09)	<0.001	1.18 (0.15)	<0.001	0.70 (0.11)	<0.001	1.74 (0.21)	<0.001	0.94 (0.15)	<0.001
More than high school	1.40 (0.14)	<0.001	1.22 (0.11)	<0.001	1.25 (0.16)	<0.001	1.03 (0.13)	<0.001	1.72 (0.24)	<0.001	1.58 (0.19)	<0.001
Marital status												
Never	Ref		Ref		Ref		Ref		Ref		Ref	
Married not living with spouse	0.27 (0.45)	0.552	−0.13 (0.35)	0.705	0.34 (0.47)	0.465	−0.14 (0.38)	0.718	1.63 (1.49)	0.274	0.10 (1.12)	0.928
Married with spouse present	0.84 (0.37)	0.025	−0.14 (0.28)	0.611	0.8 (0.37)	0.032	−0.14 (0.29)	0.625	2.42 (1.41)	0.086	0.03 (1.06)	0.979
Separated/divorced/widowed	0.59 (0.38)	0.120	−0.09 (0.29)	0.746	0.75 (0.39)	0.051	−0.03 (0.31)	0.924	2.06 (1.41)	0.144	0.06 (1.06)	0.952
Residence												
Rural	Ref		Ref		Ref		Ref		Ref		Ref	
Urban	0.73 (0.09)	<0.001	0.55 (0.07)	<0.001	0.44 (0.13)	0.001	0.63 (0.11)	<0.001	0.92 (0.14)	<0.001	0.45 (0.1)	<0.001
Insurance												
No	Ref		Ref		Ref		Ref		Ref		Ref	
Yes	0.25 (0.15)	0.088	0.23 (0.11)	0.037	0.11 (0.2)	0.591	0.24 (0.16)	0.148	0.40 (0.21)	0.059	0.24 (0.15)	0.111
Physical activity												
Insufficient	Ref		Ref		Ref		Ref		Ref		Ref	
Light	0.12 (0.10)	0.228	0.02 (0.08)	0.821	0.02 (0.14)	0.897	−0.01 (0.11)	0.937	0.23 (0.14)	0.117	0.02 (0.11)	0.856
Moderate	0.04 (0.11)	0.702	0.19 (0.08)	0.019	0.25 (0.15)	0.100	0.20 (0.12)	0.085	−0.16 (0.16)	0.337	0.17 (0.11)	0.136
Vigorous	0.01 (0.14)	0.954	−0.07 (0.10)	0.437	−0.07 (0.16)	0.678	−0.08 (0.12)	0.497	0.19 (0.24)	0.443	−0.07 (0.16)	0.643
Daytime napping												
No	Ref		Ref		Ref		Ref		Ref		Ref	
Yes	0.06 (0.07)	0.431	0.06 (0.05)	0.294	0.16 (0.1)	0.111	0.03 (0.08)	0.704	−0.07 (0.11)	0.500	0.08 (0.08)	0.297
Depression												
No	Ref		Ref		Ref		Ref		Ref		Ref	
Yes	−0.40 (0.08)	<0.001	−0.28 (0.06)	<0.001	−0.41 (0.11)	<0.001	−0.31 (0.09)	<0.001	−0.4 (0.11)	<0.001	−0.26 (0.08)	0.001
Self-rated health												
Good	Ref		Ref		Ref		Ref		Ref		Ref	
Fair/Poor	−0.06 (0.09)	0.536	−0.18 (0.07)	0.006	−0.1 (0.11)	0.375	−0.22 (0.09)	0.012	−0.03 (0.14)	0.815	−0.12 (0.10)	0.237
Employment status												
Retired	Ref		Ref		Ref		Ref		Ref		Ref	
Farm	−0.01 (0.09)	0.946	0.18 (0.07)	0.006	0.04 (0.12)	0.763	0.19 (0.09)	0.047	−0.11 (0.14)	0.434	0.20 (0.10)	0.041
Work	0.29 (0.20)	0.145	−0.12 (0.15)	0.423	0.41 (0.25)	0.096	0.05 (0.20)	0.820	0.20 (0.32)	0.529	−0.29 (0.23)	0.197

Note: Bolded point estimates indicate statistical significance at *p* < 0.05; Ref = reference group. * All models were adjusted for memory-related disease, drinking status and smoking status. All of them were not statistically significant.

## Data Availability

All the data we used have been publicly released on the CHARLS website: http://charls.pku.edu.cn/pages/data/2011-charls-wave1/zh-cn.html (accessed on 14 July 2021).

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
