# Peer review of "Sex, Sleep Duration, and the Association of Cognition: Findings from the China Health and Retirement Longitudinal Study"

_ijerph, 2021, doi:10.3390/ijerph181910140_

Round 1

Reviewer 1 Report

This paper argues an important paradox that has been discussing many years in the sleep works of literature, which is about the gender difference in sleep and how that evolves with aging. 

There is abundant literature dealing with sleep differences among the aging population. In contrast to existing literature, this study lack prior discussions dealing with these issues. The introduction and theoretical background are very short and the author's discussion and conclusion are weak. I do not find the new findings or insights from this paper.

This study utilizes the longitudinal data set but it only used the year 2011 in cross-sectional design. Why? it could have been better to create longitudinal analysis to see the causal effect of poor sleep duration and mental/cognitive outcomes.

I was wondering why several important covarients are not included I the model. Such factors are living arrangements, physical limitation (ADL/IADL), employment status (retired or work), which are missing in the model. These factors are very important variables the affect sleep or cognitive/ mental status, in the prior literature. 

Also, the factor composition between males and females is not so much different in the result, but the sex difference was the main argument and is stated as a title of this research. Why?

Moreover, why only the length of night sleep was tested as an independent variable? What is the logic that the authors used only the length of night sleep, and day sleep was treated as a control variable in this study? It needs an explanation.  

Author Response

Please allow me on behalf of my coauthors to thank you for providing the feedback on our original manuscript by the title “Sex, Sleep Duration, and the Association of Cognition: China Health and Retirement Longitudinal Study” We have already addressed all the comments from editors and reviewers as follows.

  1. This paper argues an important paradox that has been discussing many years in the sleep works of literature, which is about the gender difference in sleep and how that evolves with aging. There is abundant literature dealing with sleep differences among the aging population. In contrast to existing literature, this study lack prior discussions dealing with these issues. The introduction and theoretical background are very short and the author's discussion and conclusion are weak. I do not find the new findings or insights from this paper.

Reply: We thank the reviewer 1 for his/her comments. We would like to clarify that this study is mainly discussing sleep duration and cognition (as the titled showed) rather than sleep differences among the aging population. Those are part of the introduction but not the entire content of the introduction. We agreed that there is previous evidence regarding sleep duration and cognition as well as gender difference in sleep duration. However, our study added novel evidence of gender difference on the relationship between sleep duration and cognition. Most importantly, we find that gender play a critical role in sleep-cognition relationship among aging Chinese, which is firstly reported. Additionally, although the relationship between sleep and cognition has been reported intensively, few of them address this issue among Chinese, which is about 1/6 of the entitle population in the world. Lastly, we agreed with reviewer that there is not a convinced conclusion on the issue, while this is the motivation and implication of the presented study. As for the theoretical background and discussion, we have added the relevant content accordingly as follows.

Introduction: Accordingly, the aspect of cognitive function and sleep patterns are slightly different among men and women, while the results are inconsistent. For instance, a study found that the sleep loss could impair working memory in women but not in men. The possible explanation is that compared to men, women exhibited greater signal intensity changes when performing a working memory task and were more vulnerable to sleep loss. However, another study suggested that certain aspects of cognition (e.g., sustained attention) may be more affected by sleep loss (38 h of sleep deprivation) in men than in women due to the sex differences in brain organization and anatomy.

Discussion: Relationship between sleep and cognition is complex and poorly understood. Most past sleep studies were conducted in laboratories with strict conditions. Long sleep duration was associated with better cognitive function among children, while this relationship was not clear among elderly. A previous study pointed out that both short and long sleep duration could be associated with worse cognitive outcomes, as indicated by a U-shape or non-linear relationship between sleep duration and cognition. The latter study also demonstrated that sleep duration might have a small negative effect on reaction time and visual memory. Moreover, according to a national study of US older adults, long sleep duration may be a marker of fragmented sleep or neurodegeneration. Our study observed a significant relationship between long sleep duration and worse performance on cognition, although the association between short sleep duration and the cognition outcomes (i.e., mental health and memory decline) are not statistically significant. The biological mechanism could be listed as follows. First, sleep duration has been potentially linked with sleep quality. Poor sleep quality or sleep deprivation can disrupt the circadian rhythm regulated by gene expression in the frontal, thalamic, hypothalamic, and the locus coeruleus areas of the brain, causing a disturbance in neurogenesis and hippocampal function, then resulting in cognitive decline. Sleep-related disorders may have different effects on brain function linked to specific cognitive domains such as the synchronization function of the prefrontal cortex and neuromodulator system in visual memory. Second, the relationship between sleep and cognitive function could be partly explained by hormone melatonin which plays a critical role in sleep and cognitive performance especially among the elderly. Further, the alternative explanation is circadian factors interact to influence cognitive performance. Long sleep duration may contribute circadian disruptions linked with sleep dysregulation and impaired cognition. Therefore, long sleep duration may represent a potential causal pathway for cognitive dysfunction.

  1. This study utilizes the longitudinal data set but it only used the year 2011 in cross-sectional design. Why? it could have been better to create longitudinal analysis to see the causal effect of poor sleep duration and mental/cognitive outcomes.

Reply: Thank you for pointing out this concern. We agree that a longitudinal design can help us see the causality between the exposure and outcomes and that is exactly our next plan. We used a cross-sectional design to first establish whether there are links or associations between sleep duration and cognition. We would like to have further study using a longitudinal study design to study the cause and effect in the future. However, the follow up data are still in the data cleaning stage and will be available in the coming future. Hopefully we will have our updated results in the next coming paper. We have added this in the Discussion and Conclusions.

Discussion: First, our findings were based on a cross-sectional design and causality cannot be established between sleep duration and cognition.

Conclusions: Further longitudinal studies are warranted in establishing the causality regarding the sleep-cognition relationship.

  1. I was wondering why several important covarients are not included I the model. Such factors are living arrangements, physical limitation (ADL/IADL), employment status (retired or work), which are missing in the model. These factors are very important variables the affect sleep or cognitive/ mental status, in the prior literature. 

Reply: Thank you for your important comments. We included the important variables that suggested by previous literature [3-5] and also considered the variables you put forward. However, living arrangement information is not available in our data. For ADL/IADL, we do see many studies suggested that they are associated with cognition, but normally they are considered as results of executive dysfunction or cognition impairment, rather than the risk or protective factors in our study. Hence, we decided to not include them in our covariates. Maybe in our future studies, we could investigate ADL/IADL as outcomes. We included the employments status in the model as you suggested, while it does not fundamental change the results. Please find the updated results in Tables.

References

  1. Henry, A.; Katsoulis, M.; Masi, S.; Fatemifar, G.; Denaxas, S.; Acosta, D.; Garfield, V.; Dale, C. E. The relationship between sleep duration, cognition and dementia: a Mendelian randomization study. Int J Epidemiol 2019, 48, 849-860, doi: 10.1093/ije/dyz071.
  2. Sabia, S.; Fayosse, A.; Dumurgier, J.; van Hees, V. T.; Paquet, C.; Sommerlad, A.; Kivimäki, M.; Dugravot, A.; Singh-Manoux, A. Association of sleep duration in middle and old age with incidence of dementia. Nat. Commun. 2021, 12, 2289, doi: 10.1038/s41467-021-22354-2.
  3. Ma, Y.; Liang, L.; Zheng, F.; Shi, L.; Zhong, B.; Xie, W. Association Between Sleep Duration and Cognitive Decline. JAMA Netw Open 2020, 3, e2013573-e2013573, doi: 10.1001/jamanetworkopen.2020.13573.

  1. Also, the factor composition between males and females is not so much different in the result, but the sex difference was the main argument and is stated as a title of this research. Why?

Reply: Thank you for your feedback. The results showed not much difference in the mental status score between males and females, while there is a difference in the memory score between males and females. In the final model, only men with long sleep have been reported to have a low cognition. It is a clear clue shows that women with long sleep was not associated with low cognization, which is totally different the direction with the result from men. We expanded the main Discussion other than gender differences as follows.

Relationship between sleep and cognition is complex and poorly understood. Most past sleep studies were conducted in laboratories with strict conditions. Long sleep duration was associated with better cognitive function among children, while this relationship was not clear among elderly. A previous study pointed out that both short and long sleep duration could be associated with worse cognitive outcomes, as indicated by a U-shape or non-linear relationship between sleep duration and cognition. The latter study also demonstrated that sleep duration might have a small negative effect on reaction time and visual memory. Moreover, according to a national study of US older adults, long sleep duration may be a marker of fragmented sleep or neurodegeneration. Our study observed a significant relationship between long sleep duration and worse performance on cognition, although the association between short sleep duration and the cognition outcomes (i.e., mental health and memory decline) are not statistically significant. The biological mechanism could be listed as follows. First, sleep duration has been potentially linked with sleep quality. Poor sleep quality or sleep deprivation can disrupt the circadian rhythm regulated by gene expression in the frontal, thalamic, hypothalamic, and the locus coeruleus areas of the brain, causing a disturbance in neurogenesis and hippocampal function, then resulting in cognitive decline. Sleep-related disorders may have different effects on brain function linked to specific cognitive domains such as the synchronization function of the prefrontal cortex and neuromodulator system in visual memory. Second, the relationship between sleep and cognitive function could be partly explained by hormone melatonin which plays a critical role in sleep and cognitive performance especially among the elderly. Further, the alternative explanation is circadian factors interact to influence cognitive performance. Long sleep duration may contribute circadian disruptions linked with sleep dysregulation and impaired cognition. Therefore, long sleep duration may represent a potential causal pathway for cognitive dysfunction.

  1. Moreover, why only the length of night sleep was tested as an independent variable? What is the logic that the authors used only the length of night sleep, and day sleep was treated as a control variable in this study? It needs an explanation. 

Reply: Thank you for your concern. I consider that when reviewer says, “day sleep”, he/she means “daytime nap”. Based on culture difference and most existing western literature, when assessing the sleep duration, they usually refer to only nighttime sleep duration. However, we appreciate reviewer’s suggestion and shall have another paper talking about the total sleep duration (nighttime + daytime) and cognition (also, if possible, we will conduct a longitudinal design), and we believe we can find something interesting in the future.

Reviewer 2 Report

Reviewer comments and suggestions

This study observed the association between sleep duration and cognitive impairment (mental status and memory) among elderly Chinese. The study was longitudinal China Health and Retirement Longitudinal Study 

(CHARLS) collected baseline data for adults aged 65 years and over (N=4,785) in 2011. 

The study result noted that 59.7% of the participants had short sleep duration (<7 hours/day), and 9.0% of them had long sleep duration (>8 hours/day). Compared to normal sleep duration, long sleep duration was associated with lower mental status scores (β=-0.43, p=0.001) and lower memory scores (β=-0.27, p=0.005). The study highlighted the result of long sleep duration which was significantly associated with poorer mental status and memory scores in elderly Chinese. 

Below are the comments to be incorporated into the manuscript

  1. Line 40 The author can mention some reference
  2. Line 55-56 please explore the mechanism for these difference
  3. Line 80-81 Ethical approval number should be present here
  4. Line 91-93 is this method was previously proposed
  5. As the population was elderly, so the standard criteria of sleep duration are not same for all generalized population
  6. Line 168-169 The lines need an explanation for the outcome
  7. Line 181 Has the author checked these parameters? if not then how come they state here about fragmentation
  8. Line 214 Table?
  9. Line 218-219 The explanation was not clear, please elaborate
  10. Line 229-230 Which table was suggesting these results, please write here
  11. Most of the references used in the study was not following MDPI guidelines, please check all and correct it

Author Response

Response

Please allow me on behalf of my coauthors to thank you for providing the feedback on our original manuscript by the title “Sex, Sleep Duration, and the Association of Cognition: China Health and Retirement Longitudinal Study” We have already addressed all the comments from editors and reviewers as follows.

This study observed the association between sleep duration and cognitive impairment (mental status and memory) among elderly Chinese. The study was longitudinal China Health and Retirement Longitudinal Study (CHARLS) collected baseline data for adults aged 65 years and over (N=4,785) in 2011. 

The study result noted that 59.7% of the participants had short sleep duration (<7 hours/day), and 9.0% of them had long sleep duration (>8 hours/day). Compared to normal sleep duration, long sleep duration was associated with lower mental status scores (β=-0.43, p=0.001) and lower memory scores (β=-0.27, p=0.005). The study highlighted the result of long sleep duration which was significantly associated with poorer mental status and memory scores in elderly Chinese. 

Below are the comments to be incorporated into the manuscript

1. Line 40 The author can mention some reference

Reply: Thank you for your suggestion. We have added two references accordingly.

2. Line 55-56 please explore the mechanism for these difference

Reply: Thank you for your comments. We have revised the content accordingly.

3. Line 80-81 Ethical approval number should be present here

Reply: Thank you very much for your suggestion. We included the information in the acknowledgment part, but we added the relevant content again accordingly.

4. Line 91-93 is this method was previously proposed

Reply: Yes. The memory score and the mental status are actually from the same literature, but we only cited once. We cited the reference again accordingly. Thanks.

5. As the population was elderly, so the standard criteria of sleep duration are not same for all generalized population

Reply: Yes. That is why we choose the elder Chinese who are aged 65 years old or older and our results are generalizable to these aged Chinese population. We explained this in the method part and also added this in the limitation part.

6. Line 168-169 The lines need an explanation for the outcome

Reply: Thank you for your feedback. I guess when you say, “an explanation for the outcome”, you are referring to the cognition. Actually, the whole following paragraph is explaining the sleep duration and outcomes. We have been extended a little bit more regarding the outcome.  

7. Line 181 Has the author checked these parameters? if not then how come they state here about fragmentation

Reply: Thank you for your feedback. We mentioned fragmentation just wanted to put forward a potential hypothesis that sleep quality may influence the cognitive outcomes. In our study, sleep duration served as one aspect of sleep quality though sleep quality included many aspects such as sleep disturbance, etc. We included this in our limitation part since sleep quality is not accessible in CHARLS data.

8. Line 214 Table?

Reply: Table 2, marital status, β=0.84, p=0.024.

9. Line 218-219 The explanation was not clear, please elaborate

Reply: Thank you for your suggestion. We have already revised the sentence.

10. Line 229-230 Which table was suggesting these results, please write here

Reply: Table 2 as well.

11. Most of the references used in the study was not following MDPI guidelines, please check all and correct it

Reply: Thank you for pointing out this. We did not notice it since we download IJERPH endnote style online directly and applied it for all references. We have already adjusted them accordingly.

Reviewer 3 Report

The authors should indicate in the abstract the mean age of the participants and the percentage of men and women. They should also indicate which method they used to measure mental status and memory.

"Long sleep duration was significantly associated with poorer mental status and memory scores in elderly Chinese". This is a categorical statement that cannot be extracted from an isolated study. Rephrase the sentence specifying that these are data from this study.

Stratified by sleep duration groups are very heterogeneous. There are large differences in the percentages of participants, as well as in factors that may influence mental status, such as age, education, marital status, smoking, physical activity, depression, self-assessed health…. Being the sample is so large, if the effect of memory status is related to sleep duration, as stated by the author, it should not be lost in the multivariate model.

In the methods, it would be convenient that the multivariate model would be also adjusted for age.

The use of some categorical phrases in the discussion is not recommended, as certain conclusions cannot be obtained from an isolated study. I am refering to statements such as: "The results of our study indicate that prolonged sleep duration was associated with poorer cognitive performance as measured by mental status scores and memory scores, compared to normal sleep duration."

Author Response

Response

Please allow me on behalf of my coauthors to thank you for providing the feedback on our original manuscript by the title “Sex, Sleep Duration, and the Association of Cognition: China Health and Retirement Longitudinal Study” We have already addressed all the comments from editors and reviewers as follows.

  1. The authors should indicate in the abstract the mean age of the participants and the percentage of men and women. They should also indicate which method they used to measure mental status and memory.

Reply: Thank you for your suggestions. To be honest, we included this information at the beginning. However, IJERPH limited abstract to 200 words so we deleted them after. Now we have retrieved them again. 

  1. "Long sleep duration was significantly associated with poorer mental status and memory scores in elderly Chinese". This is a categorical statement that cannot be extracted from an isolated study. Rephrase the sentence specifying that these are data from this study.

Reply: Thank you for your suggestions and we have already rephrased the sentence.

  1. Stratified by sleep duration groups are very heterogeneous. There are large differences in the percentages of participants, as well as in factors that may influence mental status, such as age, education, marital status, smoking, physical activity, depression, self-assessed health…. Being the sample is so large, if the effect of memory status is related to sleep duration, as stated by the author, it should not be lost in the multivariate model. In the methods, it would be convenient that the multivariate model would be also adjusted for age.

Reply: Thank you for your comments. We did not stratify the model by sleep duration but by gender. We adjusted for age, education and other covariates as you mentioned in the multivariable model to address the differences you mentioned above. We may need your further clarification by “lost in the multivariate model” because we do see significant result for the association between memory score and sleep duration in the multivariable models, among both the overall sample and the males.    

  1. The use of some categorical phrases in the discussion is not recommended, as certain conclusions cannot be obtained from an isolated study. I am refering to statements such as: "The results of our study indicate that prolonged sleep duration was associated with poorer cognitive performance as measured by mental status scores and memory scores, compared to normal sleep duration."

Reply: Thank you and we have already rephrased the relevant sentences.

Reviewer 4 Report

The authors of the manuscript entitled “Sex, Sleep Duration, and the Association of Cognition: China Health and Retirement Longitudinal Study” aims to examine the association between sleep duration and cognitive impairment among elderly Chinese.

General comment:

This is an interesting research using a large sample size, however the study has important limitations. Some suggestions or questions to improve the quality of the work are presented:

Introduction

In the objective it should be stated that a gender analysis is one of the objectives of the study.

Methods

How Physical activity was assessed?  Please specify.

Discussion

Considering the limitations of the study, lines 247-252 should be considered with caution. This should be reported

Taking into account that the cognitive function mean is 6.50 points in the subjects included in the study, this should be considered as a limitation as patients with cognitive dysfunction could biased the results

Author Response

Response

Please allow me on behalf of my coauthors to thank you for providing the feedback on our original manuscript by the title “Sex, Sleep Duration, and the Association of Cognition: China Health and Retirement Longitudinal Study” We have already addressed all the comments from editors and reviewers as follows.

The authors of the manuscript entitled “Sex, Sleep Duration, and the Association of Cognition: China Health and Retirement Longitudinal Study” aims to examine the association between sleep duration and cognitive impairment among elderly Chinese.

General comment: This is an interesting research using a large sample size, however the study has important limitations. Some suggestions or questions to improve the quality of the work are presented:

Introduction

  1. In the objective it should be stated that a gender analysis is one of the objectives of the study.

Reply: Thank you so much for your suggestions. We have already adjusted the relevant content accordingly.

Methods

  1. How Physical activity was assessed?  Please specify.

Reply: Physical activity was assessed by asking “During a usual week, did you do any vigorous/moderate/light activities for at least 10 minutes continuously?” The answers were categorized into 4 levels: vigorous/moderate/light/insufficient activities. If the participants answered “No” to all of them then they were grouped into insufficient activity.

Discussion

  1. Considering the limitations of the study, lines 247-252 should be considered with caution. This should be reported.

Reply: Thank you for your comments. We rephrased the sentences by using “may” and we suggested future longitudinal studies are warranted in investigating the sleep-cognition relationship.

  1. Taking into account that the cognitive function mean is 6.50 points in the subjects included in the study, this should be considered as a limitation as patients with cognitive dysfunction could biased the results

Reply: Thank you for your valuable insights. We added it into our limitation part.

Round 2

Reviewer 1 Report

I appreciate the authors's effort to make improvements to this paper. Still, I think the title does not match the findings, and the literature review and the discussion are weak for publishing. 

Author Response

Thank you for the reviewer 1’s new comments. Regarding “I think the title does not match the findings”, we added the title as “Finding from” is stead of the original one since it may be confusing. Our study used baseline data and is a cross-sectional design while the original title sounds like a longitudinal design. We also changed the sentence in the method part as “this cross-sectional study…” and hope this clarifies the reviewer’s concern. Additionally, as for the introduction and discussion, we tried our best to address them in the first round. If the reviewer 1 is still not satisfied, please provide your specific suggestions/points on where and how we should change (since the comments were too general). Thank you.

Reviewer 3 Report

The article has been substantially improved, but it would be advisable to make some minimal changes before its publication.
Demographic data about the sex or the age of the participants appear in the abstract, but not in the article. It would be recommended to complete these data with the Average Age of the sample and its standard deviation.

Author Response

Thank you for the valuable suggestions. We added the overall descriptive information in the result part. As we used age as a categorical variable, we provided the percentages instead of means and standard errors/deviations.